# Effects of Extraction Solvents on the Total Phenolic Content, Total Flavonoid Content, and Antioxidant Activity in the Aerial Part of Root Vegetables

Eman A. Mohammed [1], Ismat G. Abdalla [1], Mohammed A. Alfawaz [2], Mohammed A. Mohammed [2], Salah A. Al Maiman [2], Magdi A. Osman [2], Abu ElGasim A. Yagoub [2] and Amro B. Hassan [2,*]

1   Department of Food Science and Technology, Faculty of Agriculture, University of Khartoum, Khartoum North 13314, Sudan
2   Department of Food Science and Nutrition, Collage of Food and Agricultural Sciences, King Saud University, P.O. Box 2460, Riyadh 11451, Saudi Arabia
*   Correspondence: ahassan2ks.c@ksu.edu.sa; Tel.: +966-500-204-967

**Abstract:** The present study aimed to investigate the total phenolics, total flavonoids, and antioxidant activity in terms of the DPPH scavenging, reducing power, and $H_2O_2$ scavenging of the aerial parts of onion, white radish, red radish, carrot, and beet as affected by different extraction solvents. Generally, the aerial part of these vegetables has high antioxidant properties. Samples were extracted with methanol (100 and 70%), ethanol (100 and 70%), and water. Total phenolic content was highest when the samples were extracted using 100% methanol, while extraction with 100% ethanol yielded the highest total flavonoids. The highest DPPH activity and $H_2O_2$ scavenging values were obtained by extraction of the aerial plant parts with 70% ethanol, and the 70% methanol extract had the highest reducing power. Partial least regression (PLS) was performed to validate the optimum solvent for extraction of the antioxidants and their activity in each plant. The PLS indicated that there was a variation in the validation of the different extracts for each plant. The high antioxidant capacity of root vegetables, which is natural, indicates that they may have health and dietetic advantages for consumers.

**Keywords:** aerial parts; root vegetables; extraction solvents; phenolic compounds; antioxidant activity

## 1. Introduction

Fruits and vegetables are among the main sources of daily caloric intake, and they also provide appreciable amounts of vitamins and pro-vitamins. Moreover, their benefits extend to the provision of a variety of phenolic substances to daily intake [1]. Phenolics are a group of phytochemicals that are categorized into simple phenols, phenolic acids, hydroxycinnamic acid derivatives, and flavonoids, which have antioxidant properties [1,2].

Generally, bioactive compounds extracted from plants are regarded as adequate sources of natural antioxidants with significant health benefits. Relatively large quantities of these compounds may be present in the seeds and peels of many vegetables and fruits [3]. Phenolic compounds have been associated with antimutagenic effects [4] and good antioxidant activity [4,5]. In addition, they may be used as natural additives in food manufacturing to maintain several quality characteristics of food, such as freshness and prevention of browning and rancidity, particularly in foods containing large quantities of fats or oils.

Root vegetable wastes (leaves and stems) are very rich in phenolic compounds with high antioxidant activity. Radish (*Raphanus sativus*) leaves and stems, which are usually discarded, possess a high radical scavenging activity [6]. Their leaves and stems have total polyphenolic contents of 86.16 and 78.77 mg/g dry extract, respectively. Moreover, an HPLC analysis revealed that radish stems and leaves contained several types of phenolic

compounds including catechin, protocatechuic acid, syringic acid, vanillic acid, ferulic acid, sinapic acid, o-coumaric acid, myricetin, and quercetin [6]. Accordingly, the aerial parts of root vegetables, that are normally discarded, can be considered as good sources of natural antioxidants in the functional food system, due to their considerable amounts of polyphenolics [6].

For the analysis of the bioactive constituents of plant materials, extraction is the most important step in phytochemical processing. Consequently, the selection of a suitable extraction technique is critical for upscaling purposes [7,8]. Since phenolic compounds are diverse in structure [9], solvent polarity influences their solubility [10], as it may affect extraction yield and activity. Flavonoids and their glycosides are easily extracted using ethanol, whereas phenolic acids and catechin are more efficiently extracted using methanol [11,12]. Moreover, antioxidant activity is also affected by the solvent used for extraction [13]. However, it has been reported that the ideal solvent for extraction varies among food matrices and types [14,15]. For this reason, it is of importance to investigate the optimal extraction solvent for a particular sample type. Therefore, this study aimed to investigate the effect of different solvents on the extraction of phenolic compounds and flavonoids from the aerial parts of root vegetables as well as to investigate the antioxidant properties of the extracts.

## 2. Materials and Methods

### 2.1. Materials

The aerial parts of onions (*Allium cepa*), white radish (*Raphanus sativus* var. Longipinnatus), red radish (*R. sativus*), beet (*Beta vulgaris*), and carrot (*Daucus carota*) were obtained from a local market during the summer season of 2019 (Central market, Khartoum, Sudan). The samples were washed thoroughly, dried at room temperature ($30 \pm 2$ °C for 24 h), and then stored at 4 °C for further analysis. All chemicals used in this study were of analytical grade.

### 2.2. Preparation of Sample Extracts

The dried aerial parts of the root vegetables were suspended in different solvents (water, 70 and 100% methanol, and 70 and 100% ethanol) at a solid-to-solvent ratio of 1:25 ($w/v$). The mixture was stirred for 24 h, followed by filtration using Whatman No. 1 filter paper (Sigma-Aldrich, St Louis, MO, USA). The filtrate of the sample was vacuum dried and then used for the analysis of total phenolic compounds (TPC), flavonoids, and antioxidant activity.

### 2.3. Antioxidant Activity

2.3.1. DPPH Scavenging Assay

The DPPH radical scavenging ability of the extracts from the aerial parts of the root vegetables was determined following the method reported previously [16]. Approximately 1.0 mL of 0.1 M DPPH was added to 0.9 mL of a 50 mM Tris-HCl buffer (pH 7.4), and 0.1 mL of the sample extract or deionized $H_2O$, the control, were mixed and then incubated at room temperature for 30 min. After the incubation period, the absorbance of the mixture was determined at 517 nm using a UV-Vis spectrophotometer. The DPPH scavenging activity was calculated according to the following formula:

$$\text{DPPH scavenging (\%)} = ((\text{Absorbance control} - \text{Absorbance sample}))/(\text{Absorbance control}) \times 100$$

2.3.2. Ferric Reducing Antioxidant Power

The ferric reducing power of the samples was determined following the method of Gulcin, Oktay, Kufre, Vioglu, and Aslan [17]. Briefly, 2.5 mL of phosphate buffer (0.2 M, pH 6.6) and 2.5 mL of 1% potassium ferricyanide were added to the extract (1 mL). The mixtures were incubated at 50 °C for 20 min, followed by the addition of 2.5 mL of 10% trichloroacetic acid and centrifugation at 1038 g for 10 min at 20 °C $\pm$ 2. Then, 2.5 mL of the supernatant was mixed with 2.5 mL of distilled $H_2O$ and 0.5 mL of 0.1% ferric chloride.

The absorbance of the mixture was measured at 700 nm. Ascorbic acid was used as a reference standard, and the results were expressed as ascorbic acid equivalents (AAE) per gram of sample.

### 2.3.3. Hydrogen Peroxide Scavenging Assay

The hydrogen peroxide ($H_2O_2$) scavenging activity of the samples was determined according to the method described by Jayaprakasha, Jaganmohan, and Sakariah [18]. For the assay, 1 mL of the sample extract (1 mg/mL) was diluted in 3 mL of a phosphate buffer (0.2 M, pH 7.4), followed by the addition of 1 mL of 40 mM $H_2O_2$, prepared in the phosphate buffer (pH 7.4). After incubating for 10 min, the absorbance of the reaction mixture was measured at 230 nm. The $H_2O_2$ scavenging ability of the sample was calculated as follows:

$$H_2O_2 \text{ scavenging (\%)} = ((\text{Absorbance of the Control} - \text{Absorbance of the sample}))/(\text{Absorbance of the control}) \times 100$$

### 2.4. Total Phenolic Content Determination

Total phenolic content (TPC) determination was conducted using the Folin–Ciocalteu method described by Waterhouse [19]. An aliquot (20 μL) of the dried sample–extract solution, prepared in methanol (1:10, *w/v*), was mixed with 1.58 mL of distilled water and 100 μL of the Folin–Ciocalteu reagent. Then, 300 μL of the $Na_2CO_3$ solution (5%) was added to the mixture, and it was kept in the dark at 25 °C for 2 h. The absorbance of the mixture was measured at 765 nm. A blank was also prepared by using distilled water following the same procedure. The TPC was expressed as gallic acid equivalents (GAE) in milligrams per gram dry extract. Gallic acid solutions of different concentrations (5, 10, 20, 40 and 80 mg/L) were used to prepare the calibration curve ($R^2 = 0.9672$).

### 2.5. Total Flavonoids Determination

Total flavonoid content (TFC) was determined following the method reported by Kim, Jeong, and Lee [20]. Briefly, a mixture of the extract (1 mL), 5% $NaNO_2$ solution (300 μL), and 10% aluminum chloride (300 μL) was incubated at 25 °C for 5 min. Then, 1 N sodium hydroxide (2 mL) was added to the mixture. The volume of the mixture was completed to 10 mL with water and mixed thoroughly using a vortex. The absorbance was measured at 510 nm. A catechin calibration curve was prepared ($R^2 = 0.974$). The TFC of the sample was stated as mg catechin equivalents (CE)/g sample, dry basis.

### 2.6. Statistical Analysis

Samples were analyzed in triplicate, and the means were computed. Data were analyzed using one-way analysis of variance (ANOVA) [21]. Multiple significant differences in the means ($p < 0.05$) were determined using the least significant difference (LSD) range test. Linear Partial Least Squares Regression Analysis (PLS) was used to analyze the relationships between the different extracts (active variable; X) and the phytochemical contents and its antioxidant activities (Y variable), using XLSTAT software [22].

## 3. Results and Discussion

### 3.1. Antioxidant Activity of the Aerial Part of Root Vegetables

Various antioxidant activity assays such as the (DPPH) scavenging activity, ferric reducing power, and hydrogen peroxide scavenging activity assays were used to evaluate the effect of extraction solvents on the antioxidant activity in the aerial part of root vegetables.

Table 1 shows the DPPH scavenging activity of different solvent extracts from the aerial parts of onions, white radishes, red radishes, beets, and carrots. There was a significant difference in the DPPH scavenging activity among the extracts from the different root vegetables ($p < 0.05$). Irrespective of the root vegetable, aqueous and 100% ethanolic extracts had the lowest DPPH scavenging activity compared to the other solvent extracts.

**Table 1.** DPPH scavenging activity (%) of the aerial part of selected root vegetables.

| Extracts | Root Vegetables | | | | |
| | Onion | White Radish | Red Radish | Beet | Carrot |
| --- | --- | --- | --- | --- | --- |
| Aqueous | 54.67 ± 0.95 [c] | 40.46 ± 0.87 [b] | 80.48 ± 0.98 [b] | 62.98 ± 0.19 [c] | 64.31 ± 0.19 [c] |
| 70% Methanol | 67.60 ± 0.52 [b] | 81.59 ± 1.42 [a] | 85.84 ± 1.64 [a] | 62.07 ± 1.66 [c] | 82.53 ± 0.47 [a] |
| 100% Methanol | 68.51 ± 0.90 [b] | 77.81 ± 1.64 [a] | 76.86 ± 0.94 [c] | 82.22 ± 0.98 [a] | 74.82 ± 1.19 [b] |
| 70% Ethanol | 75.99 ± 0.58 [a] | 81.90 ± 1.91 [a] | 81.43 ± 0.98 [b] | 72.14 ± 2.17 [b] | 82.22 ± 0.72 [a] |
| 100% Ethanol | 56.25 ± 1.19 [c] | 42.85 ± 1.16 [b] | 54.41 ± 0.81 [d] | 62.02 ± 2.29 [c] | 55.15 ± 1.42 [d] |
| F-test | ** | ** | ** | ** | ** |
| LSD 0.05 | 4.09 | 4.36 | 2.63 | 3.52 | 1.62 |

Values are means ± SD. Values not sharing a similar superscript in a column are significantly different ** different at ($p < 0.05$) as assessed by LSD.

The DPPH radical scavenging activity of the different solvent extracts from the aerial part of onions ranged between 54.67 and 75.99 %, with a descending manner of activity as follows: 70% ethanol > 100% methanol > 70% methanol > 100% ethanol > water. For white radish, the DPPH radical scavenging activity of the solvent extracts ranged from 40.46% to 81.90%. The 70% methanolic extract had a significantly ($p < 0.05$) higher DPPH radical scavenging activity (85.84%), followed by the 70% ethanolic extract (81.43%), the aqueous extract (80.49%), the 100% methanolic extract (76.86%), and lastly the 100 ethanolic extract (54.41%). For beet, the aqueous, 70% methanolic, and 100% ethanolic extracts showed the lowest DPPH radical scavenging activities (62.98, 62.07, and 62.02%, respectively). However, the extraction of the samples with 100% methanol resulted in a significantly higher DPPH radical scavenging activity of 82.22% ($p < 0.05$). When 70% methanol was used, the carrot extract exhibited significantly higher scavenging activity (82.53%, $p < 0.05$) which was followed by 70% ethanol and 100% methanol (82.22 and 74.82%, respectively).

The higher DPPH radical scavenging activity of the onion aerial parts was accompanied by a higher level of phenolic content and lower level of flavonoids (Table 1) which may indicate a greater contribution of phenolics to antioxidant activity. All extracts obtained using organic solvents gave a stronger radical scavenging capacity than that of the water extract except the 100% ethanolic extract. A similar trend was observed in a study of the DPPH radical scavenging activity of pineapple crude extract [23] and defatted wheat germ [24]. Karadeniz, Burdurlu, Koca, and Soyer [25] reported a 22.5% antioxidant activity of onion, which was lower than the activity observed in the present study, but in agreement with our findings, they concluded that the aerial parts of root vegetables had better antioxidant properties compared to the bulb.

The DPPH radical scavenging activity of the red radish extracts decreased with increasing water content in aqueous solvents. The higher DPPH radical scavenging activity of the white and red radish aerial parts could be due to the higher TPC and TFC which enhanced the antioxidant activity of the samples. Radish leaves were observed to have more antioxidant activity than the roots, and their most abundant free and bound phenolic compounds were pyrogallol and vanillic acid and epicatechin and coumaric acid, respectively [26].

The high DPPH radical scavenging activity of both the aerial parts of beet and carrot could be attributed to their higher levels of phenolic which enhanced the antioxidant activity of the samples. The antioxidant activity of red beetroot was reported to vary from 14.2% to 90.7% [26,27]. Moreover, Kaur and Kapoor [28] stated that the DPPH radical scavenging activity of the ethanolic and water extracts of red beetroot was 73.3 and 55%, respectively. The TPC and antioxidant activity of red beetroot may be influenced by variety, growing and postharvest conditions, soil composition, and climatic conditions. Charanjit and Harish [28] reported that the antioxidant activities of the ethanolic and water extracts obtained from carrot root were 67.0 and 37.5%, respectively, while those of the leaves were 66.5 and 63.5%, respectively. The results revealed that the aerial parts of both beetroot and carrot are good sources of antioxidants.

Table 2 shows the ferric reducing power (mg AAE/g) of the extracts from the aerial parts of the root vegetables. The aqueous extracts from all studied samples had significantly ($p < 0.05$) lower reducing power, which was found to be 0.41, 1.65, 3.44, 4.34, and 7.13 mg AAE/g for the aerial parts of onion, white radish, red radish, beet, and carrot, respectively. On the other hand, the ethanolic (70%) and methanolic (70%) extracts from the aerial parts of onion had significantly higher ($p < 0.05$) reducing power (1.43 and 1.40 mg AAE/g, respectively) compared with the other extracts. For white radish, the 100% ethanolic extract had a significantly higher reducing power ($p < 0.05$) of 5.36 mg AEE/g. The ethanolic (70 and 100%) extracts of red radish had a significantly higher ($p < 0.05$) reducing power (6.29 and 6.13 mg AEE/g, respectively) compared to the other extracts. The reducing power of the beetroot and carrot aerial parts showed a similar trend (Table 2). Our study showed that different extracts had different reducing powers. Aqueous extracts exhibited the lowest reducing power compared to the ethanolic and methanolic extracts. A similar trend was observed in chestnut flower and cauliflower wastes [29,30]. Moreover, Do et al. [31] observed a low reducing power in water extracts of Limnophila aromatica.

**Table 2.** The ferric reducing power (mg AAE/g) of the aerial part of selected root vegetables.

| Extracts | Root Vegetables | | | | |
| --- | --- | --- | --- | --- | --- |
| | Onion | White Radish | Red Radish | Beet | Carrot |
| Aqueous | 0.41 ± 0.06 [c] | 1.65 ± 0.08 [e] | 3.44 ± 0.16 [d] | 4.34 ± 0.08 [c] | 7.13 ± 0.18 [c] |
| 70% Methanol | 1.40 ± 0.11 [a] | 4.23 ± 0.49 [b] | 4.26 ± 0.11 [b] | 9.66 ± 0.05 [a] | 10.18 ± 0.06 [a] |
| 100% Methanol | 0.45 ± 0.03 [c] | 2.65 ± 0.33 [d] | 3.99 ± 0.11 [c] | 9.68 ± 0.09 [a] | 9.88 ± 0.11 [a] |
| 70% Ethanol | 1.43 ± 0.08 [a] | 3.68 ± 0.26 [c] | 6.29 ± 0.05 [a] | 8.50 ± 0.91 [b] | 7.95 ± 0.03 [b] |
| 100% Ethanol | 1.20 ± 0.06 [b] | 5.36 ± 0.15 [a] | 6.13 ± 0.09 [a] | 8.49 ± 0.14 [b] | 7.74 ± 0.06 [b] |
| F-test | ** | ** | ** | ** | ** |
| LSD 0.05 | 0.15 | 0.53 | 0.19 | 0.17 | 0.18 |

Values are means ± SD. Values not sharing a similar superscript in a column are significantly different ** different at ($p < 0.05$) as assessed by LSD.

The hydrogen peroxide ($H_2O_2$) scavenging activity of the root vegetable wastes is shown in Table 3. In comparison to the other solvent extracts, the aqueous extracts of the aerial parts from onion, white and red radish, carrot, and beet consistently yielded significantly lower ($p < 0.05$) $H_2O_2$ scavenging activity. The ethanolic (70%) extracts of the aerial parts of all samples, except for the white radish, had a significantly higher ($p < 0.05$) $H_2O_2$ scavenging activity compared with the other solvent extracts. Most of the antioxidants exhibited concentration-dependent hydrogen peroxide scavenging activity. Similar findings were reported in fresh bokbunja (Rubus coreanus) and wine processing extracts [32].

**Table 3.** The hydrogen peroxide scavenging activity (%) of the aerial part of selected root vegetables.

| Extracts | Root Vegetables | | | | |
| --- | --- | --- | --- | --- | --- |
| | Onion | White Radish | Red Radish | Beet | Carrot |
| Aqueous | 73.34 ± 0.04 [d] | 72.98 ± 0.09 [d] | 72.38 ± 0.98 [d] | 75.13 ± 0.08 [c] | 75.70 ± 0.19 [c] |
| 70% Methanol | 79.56 ± 0.13 [b] | 83.45 ± 1.06 [a] | 76.02 ± 1.04 [c] | 83.53 ± 1.12 [a] | 82.30 ± 0.13 [a] |
| 100% Methanol | 76.06 ± 0.20 [c] | 80.63 ± 1.04 [b] | 75.33 ± 0.94 [c] | 82.48 ± 1.10 [b] | 81.12 ± 0.19 [b] |
| 70% Ethanol | 81.22 ± 0.15 [a] | 80.22 ± 1.06 [b] | 82.38 ± 1.04 [a] | 84.14 ± 1.08 [a] | 82.85 ± 0.12 [a] |
| 100% Ethanol | 79.73 ± 0.12 [b] | 78.55 ± 1.16 [c] | 80.23 ± 1.10 [b] | 83.70 ± 1.10 [a] | 82.58 ± 1.13 [a] |
| F-test | ** | ** | ** | ** | ** |
| LSD 0.05 | 0.25 | 0.11 | 0.17 | 0.17 | 0.23 |

Values are means ± SD. Values not sharing a similar superscript in a column are significantly different ** different at ($p < 0.05$) as assessed by LSD.

### 3.2. Total Phenolic Content of the Aerial Part of Root Vegetables

The TPC of the aerial part of the root vegetables is shown in Table 4. In general, the TPC varied significantly ($p < 0.05$) according to the extraction solvent used for each plant. The onion, white radish, red radish, beetroot, and carrot aerial parts demonstrated the highest TPC after extraction with 100% methanol (16.90, 29.59, 37.09, 31.73, and 66.33 mg GAE/g, respectively). The TPC of the aqueous extract was significantly lower ($p < 0.05$) than that of the other solvents used in white radish extraction (12.31 mg GAE/g), red radish (18.99 mg GAE/g), and beet (20.78 mg GAE/g). However, ethanol (100%) extraction resulted in a significantly lower ($p < 0.05$) TPC of onion (5.30 mg GAE/g) and carrot (25.30 mg GAE/g).

**Table 4.** Total phenolic content (mg GAE/g) of the aerial part of selected root vegetables.

| Extracts | Root Vegetables | | | | |
| | Onion | White Radish | Red Radish | Beet | Carrot |
|---|---|---|---|---|---|
| Aqueous | 11.01 ± 1.24 [b] | 12.31 ± 1.44 [c] | 18.99 ± 0.41 [d] | 20.78 ± 1.83 [c] | 9.59 ± 1.29 [e] |
| 70% Methanol | 11.97 ± 1.26 [b] | 30.17 ± 0.99 [a] | 36.37 ± 0.95 [ab] | 20.42 ± 1.83 [c] | 55.10 ± 1.58 [b] |
| 100% Methanol | 16.90 ± 0.65 [a] | 29.59 ± 0.36 [a] | 37.09 ± 0.36 [a] | 31.73 ± 0.95 [a] | 66.33 ± 1.49 [a] |
| 70% Ethanol | 10.90 ± 0.90 [b] | 28.83 ± 0.88 [a] | 33.24 ± 0.48 [b] | 24.23 ± 1.99 [b] | 47.44 ± 1.78 [c] |
| 100% Ethanol | 5.30 ± 0.71 [c] | 22.44 ± 0.71 [b] | 27.59 ± 0.80 [c] | 28.47 ± 1.17 [a] | 25.30 ± 1.56 [d] |
| F-test | ** | ** | ** | ** | ** |
| LSD 0.05 | 1.75 | 3.14 | 3.30 | 3.74 | 4.90 |

Values are means ± SD. Values not sharing a similar superscript in a column are significantly different ** different at ($p < 0.05$) as assessed by LSD.

Based on the results of the TPC, the best solvent for extracting the aerial part of the root vegetables was 100% methanol. This might be a result of the enhanced solubility of nonphenolic compounds, due to the presence of the water molecules, in organic solutions. It might also be due to the high solubility of phenolic compounds in methanol [5]. Moreover, the content of nonphenolic compounds such as carbohydrates and terpenes is higher in water extracts than in other extracts. As reported by Sultana, Anwar, and Ashraf, [5] complexes of phenolics with high molecular weight compounds might be formed when phenolic compounds are extracted in methanol. Moreover, it has been reported that methanol extract had the highest polyphenol content in both carrots (250 mg/100 g) and beetroot (220 mg/100 g) pulp wastes compared to ethanol and aqueous extracts of the samples, which ranged from 67 to 110 mg/100 g [28]. Many factors govern the content of the total phenolics of root vegetables, including the agrochemical characteristics of the soil of the seeding area, climatic conditions, farming and harvesting technology, and variety. In conclusion, the results of this study prove that the aerial parts of the root vegetables are a rich source of phenolic compounds.

### 3.3. Total Flavonoid Content of Aerial Part Wastes of Root Vegetables

The TFC of the aerial parts of the root vegetable samples is shown in Table 5. The TFC was influenced by the type and nature of the extraction solvent. The highest TFC of the aerial parts of onion, white radish, red radish, beet, and carrot (16.58, 61.58, 55.33, 36.31, and 47.83 mg CE/g) was obtained after using 100% ethanol as the extraction solvent. Among all samples, the aqueous extracts from the aerial parts of onion and white radish and the 70% ethanol extract from the aerial parts of onion had the lowest TFC ($p < 0.05$).

**Table 5.** Total flavonoid content (mg CE/g) of the aerial part of selected root vegetables.

| Extracts | Root Vegetables | | | | |
| --- | --- | --- | --- | --- | --- |
| | Onion | White Radish | Red Radish | Beet | Carrot |
| Aqueous | 3.71 ± 0.49 [c] | 4.78 ± 0.48 [d] | 11.17 ± 0.58 [c] | 14.64 ± 1.05 [b] | 15.75 ± 1.25 [d] |
| 70% Methanol | 4.78 ± 0.48 [bc] | 15.09 ± 1.14 [b] | 9.50 ± 0.71 [d] | 7.83 ± 0.72 [c] | 45.89 ± 1.47 [b] |
| 100% Methanol | 15.47 ± 0.87 [a] | 11.58 ± 0.72 [c] | 12.23 ± 0.93 [c] | 11.86 ± 1.05 [b] | 39.08 ± 0.72 [c] |
| 70% Ethanol | 5.47 ± 0.48 [b] | 15.33 ± 0.72 [b] | 16.03 ± 0.72 [b] | 8.25 ± 1.25 [c] | 48.25 ± 1.25 [a] |
| 100% Ethanol | 16.58 ± 0.72 [a] | 61.58 ± 1.91 [a] | 55.33 ± 0.72 [a] | 36.31 ± 1.58 [a] | 47.83 ± 0.24 [ab] |
| F-test | ** | ** | ** | ** | ** |
| LSD 0.05 | 1.12 | 1.98 | 1.77 | 2.81 | 2.99 |

Values are means ± SD. Values not sharing a similar superscript in a column are significantly different ** different at ($p < 0.05$) as assessed by LSD.

These results indicate that the aerial parts of the selected root vegetables are rich sources of antioxidants. However, it depends on the extraction solvent used. The obtained results revealed that in some cases, as the concentration of water in ethanol or methanol decreased, the total flavonoids content of the extract increased, as observed for the 100% ethanol extract. Goyeneche et al. [26] reported that the TFC of red radish leaves was four times higher than that of the root. Additionally, Marinova, Ribarova, and Atanassova [33] reported a very low content of flavonoids in white radish roots. Regardless of the extraction solvent, the TFC of red radish leaves and white radish roots obtained in this study was higher than those reported earlier [26]. In general, the results indicated that the aerial parts of these root vegetables were rich in flavonoids and hence are considered a good source of flavonoids.

### 3.4. Partial Least Squares Regression Analysis (PLS)

In this research, Partial Least Squares regression analysis (PLS) was performed to classify the validation of the different extracts in the extraction of the phytochemical compounds and their antioxidant activity for each plant. The interactive effects of different extracts (active variables) on the TPC, TFC, DPPH, FRAP, and $H_2O_2$ (y-variables) of the aerial parts of onion, white radish, red radish, beet and carrot were observed (Figure 1a–e). According to this model, there was a variation in the valid and optimum extracts for each plant. The PLS chart for the onion aerial part (Figure 1a) exhibits the association of the 100% methanol, 70% methanol, and 70% ethanol on the higher antioxidant activities and TPC; however, the 100% methanol extract was the most valid extract for these compounds. Similarly, the PLS in Figure 1b shows that the extracts of 100% methanol, 70% methanol, and 70% ethanol were the most valid in the extraction of the compounds in the aerial parts of the white radish plant, with the highest validation for the 70% ethanol extract. Similar observations are also clearly observed in Figure 1b, which also shows that 70% ethanol was the most valid and optimum extract for the aerial part of the carrot plant. The PLS of the aerial parts of red radish and beet (Figure 1c,d, respectively) shows a similar trend in the validation and optimization of the extract solvent. In both, 70% methanol had the highest validation for the antioxidants among the other studied extracts. In general, the PLS indicated that there was a variation in the validation of the different extracts for each plant, and the most valid and optimum extracts for the aerial part of the onion, white radish, red radish, beet, and carrot were found to be 100% methanol, 70% ethanol, 70% methanol, 70% methanol, and 70% ethanol, respectively.

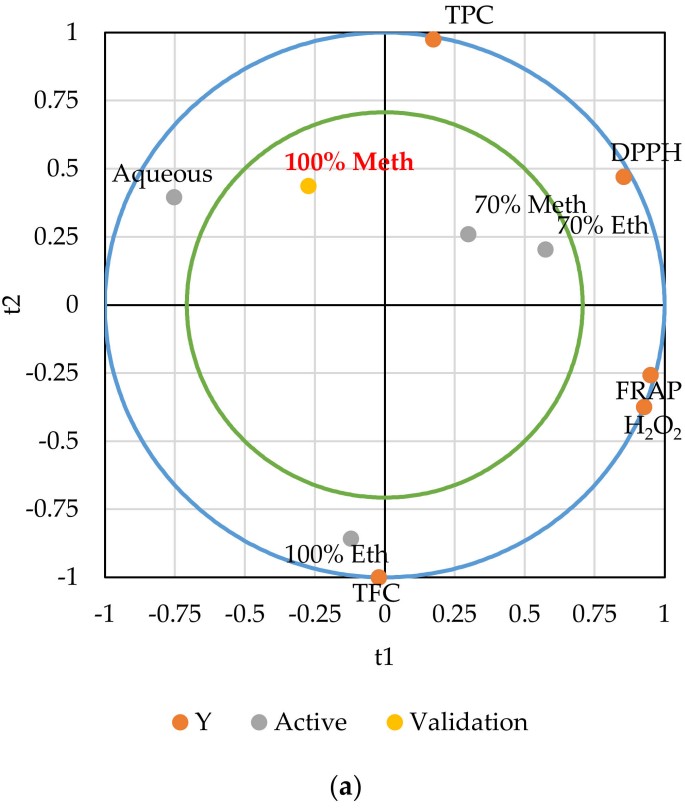

(**a**)

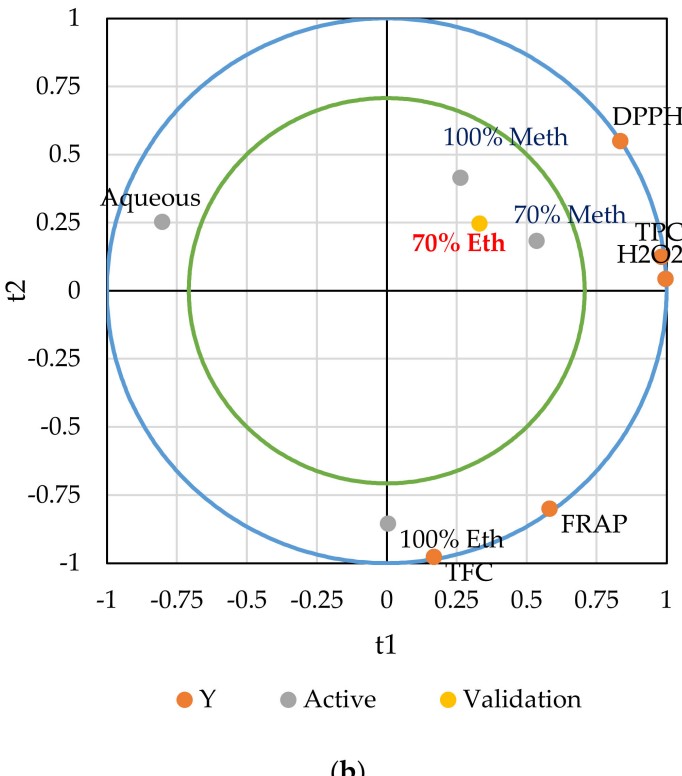

(**b**)

**Figure 1.** *Cont.*

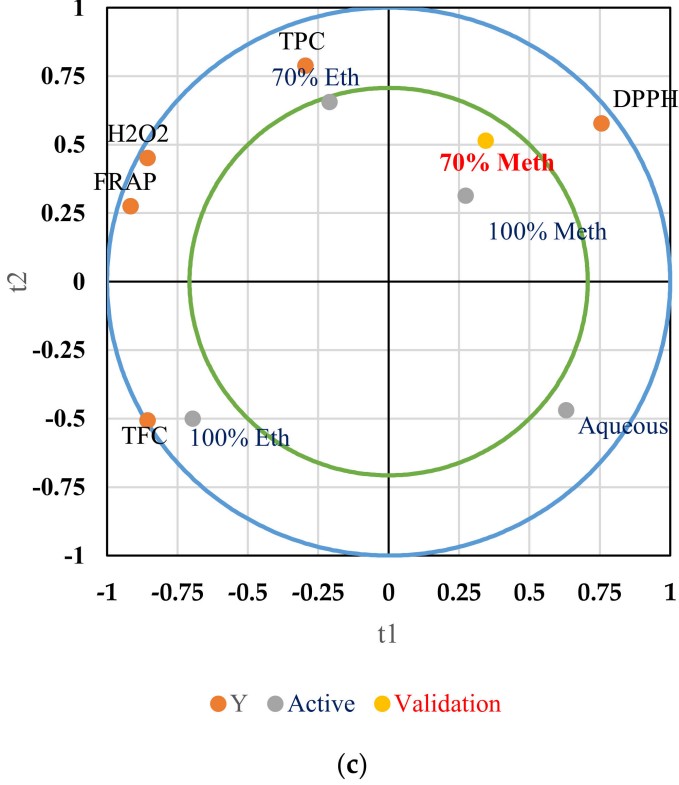

(**c**)

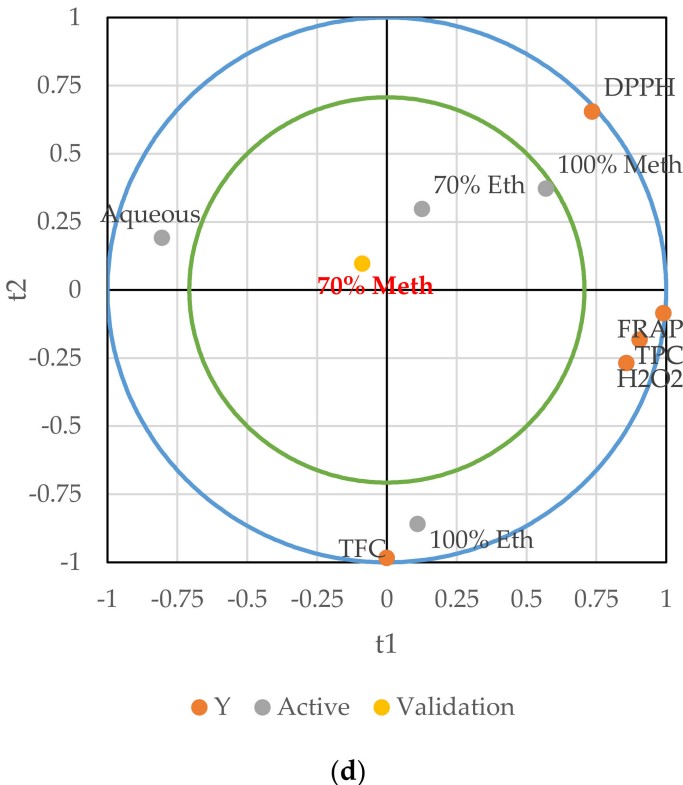

(**d**)

**Figure 1.** *Cont.*

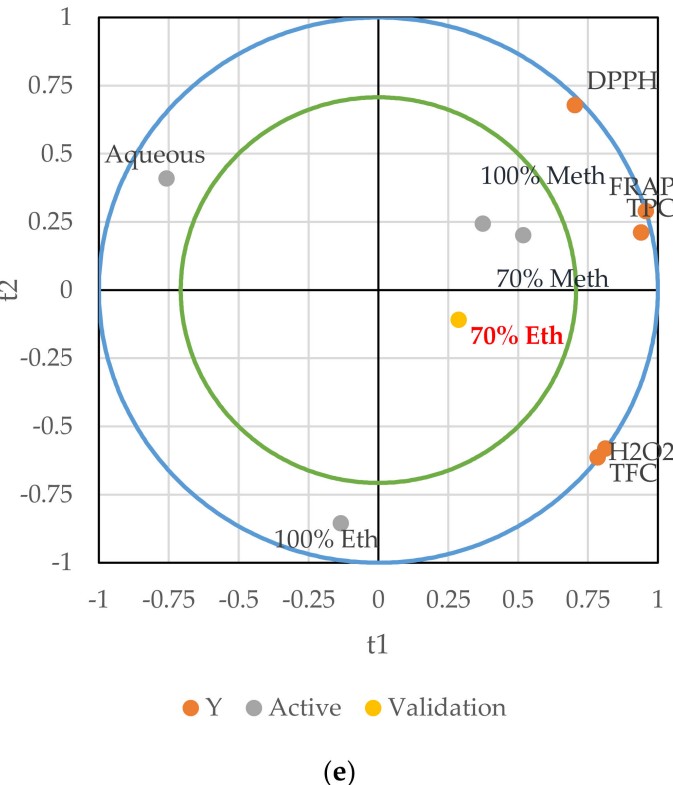

(**e**)

**Figure 1.** Partial Least Squares regression analysis (PLS) of the validation of the different extracts in the extraction of the phytochemical compounds and antioxidant activities of the aerial wasted parts of onion (**a**), white radish (**b**), red radish (**c**), beet (**d**), and carrot (**e**). Correlation on access t1 and t2.

## 4. Conclusions

In the current study, different extracts of water, 100% methanol, 70% methanol, 100% ethanol, and 70% ethanol were used to evaluate and validate their impact on the extraction of phenolic compounds and flavonoids from the aerial wasted part of the selected root vegetables. The nature of the solvent and its polarity significantly impacted the phenolic and antioxidant extraction. Their polarities ranged from polar to non-polar; hence, the validation extraction of these compounds was usually obtained in the polar solvent which had a better efficiency as a result of the interactions (hydrogen bonds) between the polar sites of the antioxidant compounds and the solvent than nonpolar ones. Consequently, the aerial parts of the selected root vegetables were found to be rich in TPC and TFC with high antioxidant activities. Moreover, the validation of these extracts varied among the different studied plants. Regardless of the extraction system, the results of this work indicated that the aerial parts of root vegetables could serve as antioxidants against free-radical-associated oxidative damage and thus can be part of the preparation of functional foods.

**Author Contributions:** E.A.M.; methodology and formal analysis, I.G.A.; validation, supervision; formal analysis, M.A.A. and M.A.M.; investigation, data curation, S.A.A.M., M.A.O. and A.E.A.Y.; writing—original draft preparation, A.B.H.; writing—review and editing, visualization project administration, funding acquisition, software. All authors have read and agreed to the published version of the manuscript.

**Funding:** Deputyship for Research & Innovation, Ministry of Education in Saudi Arabia for funding this research work through the project no. (IFKSURG-2-000).

**Institutional Review Board Statement:** Not applicable.

**Informed Consent Statement:** Not applicable.

**Data Availability Statement:** Not applicable.

**Conflicts of Interest:** The authors declare that there are no conflicts of interest.

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
