# Peer review of "Effects of Extraction Solvents on the Total Phenolic Content, Total Flavonoid Content, and Antioxidant Activity in the Aerial Part of Root Vegetables"

_agriculture, doi:10.3390/agriculture12111820_

Round 1

Reviewer 1 Report

This manuscript reported total phenolics, flavonoids, and antioxidants extracted from aerial parts of different root vegetables using different solvents. The authors also built partial least regression (PLS) was performed to validate the optimum extract in each plant.

Major concerns:

The author did all the work on the aerial parts of root vegetables such as onions and carrots. I am not sure why it is important to measure antioxidants in the non-edible parts of root vegetables. Please justify the significance of this work.

I am very confused by the PLS results plot 1. According the manuscript, each plot is for a single species and the author use different phytochemical measure as dependent variable Y. I assume the author use the results from 4 solvents as X to train the model and use the result from optimum solve to validation the model? Are grey dots in plot represent the loading of the independent variables used for training?  How does validation works? Please describe in details in the method section, especially what are the X and Y.

Minor concerns:

Please explain why partial least square regression were used instead of principal component regression.

Figure 1: It would be good to add more descriptions in the figure legend of the loading plots. E.g. Are t1 and t2 the score of the first two components what percentage of variation were explained by the two factors for X and Y respectively? Also the green and blue circles.

Author Response

Responses to the reviewer’s comments

Reviewer 1

Dear Sir:

Thank you very much for your evaluation and valuable comments on our manuscript (agriculture-1883605). In response to your kind and helpful suggestions, we have revised our manuscript and the changes we made are highlighted in red in the manuscript text. Your comments, the revisions made, and our responses to your remarks are listed below:

Reviewer 1

This manuscript reported total phenolics, flavonoids, and antioxidants extracted from aerial parts of different root vegetables using different solvents. The authors also built partial least regression (PLS) was performed to validate the optimum extract in each plant.

Major concerns:

The author did all the work on the aerial parts of root vegetables such as onions and carrots. I am not sure why it is important to measure antioxidants in the non-edible parts of root vegetables. Please justify the significance of this work.

Response:  These non-edible parts considered as one of the wasted food. Estimation the antioxidant capacity of wasted part is required, so as to explore the potentials as alternative sources of antioxidants with added value for cosmetic, pharmaceutical, or food industrial applications

https://doi.org/10.1016/B978-0-323-85253-1.00020-4

https://doi.org/10.1016/B978-0-12-813768-0.00002-5

I am very confused by the PLS results plot 1. According the manuscript, each plot is for a single species and the author use different phytochemical measure as dependent variable Y. I assume the author use the results from 4 solvents as X to train the model and use the result from optimum solve to validation the model? Are grey dots in plot represent the loading of the independent variables used for training?  How does validation works? Please describe in details in the method section, especially what are the X and Y.

Response: PLS results plot is usually used to show the valid variable among that one. PLS predict validated fitted value for each observation in the data set, the observation can be excluded from the model used to calculate the predicted response for that observation

In this study we used the phytochemical contents and its antioxidant activities (Y variable) and the different extracts (active variable; X). 

Minor concerns:

Please explain why partial least square regression were used instead of principal component regression.

 Response:  PLS is applied based on the correlation between the dependent variable and the independent variables. Also PLS technique is more efficient than the PCA technique for dimension reduction. As I mentioned formally, it usually uses to show the valid variable among the comparative variables.

Figure 1: It would be good to add more descriptions in the figure legend of the loading plots. E.g. Are t1 and t2 the score of the first two components what percentage of variation were explained by the two factors for X and Y respectively? Also the green and blue circles.

Response: t1 and t2 show the correlation matrix of the Y variables and active variable, which describes the scores corresponding to the validation set.

Thank you very much

Sincerely Yours

Dr. Amro B. Hassan

The Corresponding author

Reviewer 2 Report

The manuscript is a simple study that investigated the effects of different extraction solvents on total phenolic, flavonoid and antioxidant activity of the aerial part of root vegetables. Methods used are adequate although not extensive. Phenolic compounds were not further characterized. Also, discussion of results obtained can be improved.

Suggested comments for correction:

Lines 13, 18, 84,94, 104, 111, 203, 206, 207, 267 etc.: eg. numbers in chemical formula should be written as subscript, correct throughout manuscript; eg. H2O2

Lines 66-67: Italicize botanical names

Line 69: Indicate time duration for drying at room temperature

Line 128: P should be written in small case (p); correct throughout manuscript

Line 127: Indicate type of ANOVA, One way or Two way?  It is also not clear whether authors are considering statistical differences between extraction solvents or differences between different vegetables or both; An appropriate statistical analytical method which considers both should be used.

Line 133-182 (Antioxidant activity)

Provide scientific basis to explain how different solvents influenced DPPH activities; For-instance water extract of Red Radish gave significantly higher DPPH scavenging activity than 100% methanol and 100% ethanol

Different root vegetables have been used and this can influence antioxidant properties. Values observed are not attributable to only solvent effect.

Also compare the DPPH activities of the different root vegetables under each extraction condition.

Line 156-158;

The statement that "All extracts obtained by using organic solvents gave stronger radical scavenging capacity than that of the water extract except the 100% ethanolic extract" is not wholly true; Consider the case of 100% methanolic extract from Red Radish.

Line 186-188: Reconsider statement or statistics again; Are you statistically comparing the different root vegetables, since footnote of table 2 indicate comparison between solvents.

Again explain why different solvents are demonstrating different ferric reducing abilities

Line 204-205: Are you statistically comparing the different root vegetables or extraction solvents?

For each parameter studied also consider discussing the effect of the different vegetables alongside solvent effect. Provide scientific basis for how different solvent influenced each parameter studied.

Author Response

Responses to the reviewer’s comments

Reviewer 2

Dear Sir:

Thank you very much for your evaluation and valuable comments on our manuscript (agriculture-1883605). In response to your kind and helpful suggestions, we have revised our manuscript and the changes we made are highlighted in red in the manuscript text. Your comments, the revisions made, and our responses to your remarks are listed below:

The manuscript is a simple study that investigated the effects of different extraction solvents on total phenolic, flavonoid and antioxidant activity of the aerial part of root vegetables. Methods used are adequate although not extensive. Phenolic compounds were not further characterized. Also, discussion of results obtained can be improved.

Suggested comments for correction:

Lines 13, 18, 84,94, 104, 111, 203, 206, 207, 267 etc.: eg. numbers in chemical formula should be written as subscript, correct throughout manuscript; eg. H2O2

Response:  Done

Lines 66-67: Italicize botanical names

Response:  Done

Line 69: Indicate time duration for drying at room temperature

Response:  Done

Line 128: P should be written in small case (p); correct throughout manuscript

Response:  Done

Line 127: Indicate type of ANOVA, One way or Two way?  It is also not clear whether authors are considering statistical differences between extraction solvents or differences between different vegetables or both; An appropriate statistical analytical method which considers both should be used.

Response:  statements were added to the section.

Line 133-182 (Antioxidant activity)

Provide scientific basis to explain how different solvents influenced DPPH activities; For-instance water extract of Red Radish gave significantly higher DPPH scavenging activity than 100% methanol and 100% ethanol

Different root vegetables have been used and this can influence antioxidant properties. Values observed are not attributable to only solvent effect. Also compare the DPPH activities of the different root vegetables under each extraction condition.

Line 156-158; The statement that "All extracts obtained by using organic solvents gave stronger radical scavenging capacity than that of the water extract except the 100% ethanolic extract" is not wholly true; Consider the case of 100% methanolic extract from Red Radish.

Response:  corrected. Observations regarding this section was corrected in the text

Line 186-188: Reconsider statement or statistics again; Are you statistically comparing the different root vegetables, since footnote of table 2 indicate comparison between solvents.

Response:  in this study we aimed to investigate the effect of different solvent on the antioxidant capacity of each plant

Again explain why different solvents are demonstrating different ferric reducing abilities

Response:  each solvent has its own kinetics mechanism against the food component. (doi: 10.1007/s10068-020-00874-9  )

Line 204-205: Are you statistically comparing the different root vegetables or extraction solvents?

Response:  in this study we aimed to investigate the effect of different solvent on the antioxidant capacity of each plant

For each parameter studied also consider discussing the effect of the different vegetables alongside solvent effect. Provide scientific basis for how different solvent influenced each parameter studied.

Response:  Statement was  added.

Sincerely Yours

Dr. Amro B. Hassan

The Corresponding author

Reviewer 3 Report

Hassan et al. reported the effects of extraction solvents on total phenolic content, total flavonoid content, and antioxidant activity in the aerial part of root vegetables. This work is interesting and it would be better after the following issues are solved.

1.      What is the innovation of this article?

2.      When the dried aerial parts of the root vegetables are treated with ethanol and methanol, whether the solvent removal step is included, and whether it is safe to add organic solvents?

3.       Line 80-85, as for the DPPH scavenging assay, the number of mL of DPPH solution added to the samples and the concentration of DPPH solution are not mentioned. It is suggested to supplement and improve DPPH experimental method.

4.      As for the DPPH scavenging assay, what is the purpose of adding a 50 mm Tris-HCl buffer (pH 7.4)?

5.       Line 90-92, “Briefly, 2.5 mL of phosphate buffer (0.2 M, pH 6.6) and 2.5 mL of 1% potassium ferricyanide were added to different concentrations of extracts (5, 10, 20, 30, and 40 μg/mL)”. How many mL of extract samples with different concentrations? Please add in the text.

6.      Line 101, “a phosphate buffer (pH 7.4)”. What is the phosphate buffer concentration?

7.       Line 120-121, “1 N sodium hydroxide (2 mL) was added to the mixture”. What is 1 N?

8.       Line 141-143, “with a descending manner of activity as follows: 70% ethanol > 100% methanol > 70% methanol > 100% ethanol > water.” In terms of significance, 100% methanol and 70% methanol, 100% ethanol and water were not significant.

9.       Line 145-146, “The 70% methanolic extract had a significantly (p < 0.05) higher DPPH radical scavenging activity (85.84%), followed by the 70% ethanolic extract (81.43%), 100% ethanolic extract (54.41%), and the aqueous extract (40.46%).” By observing Table 1, except that the aqueous extract (40.46%) is the data of white radish, others are the data of red radish. Please recheck the accuracy of the data.

10.   Line 151-153, “When 70% methanol was used, carrot extract exhibited significantly higher scavenging activity (82.53%, P < 0.05) which was followed by 70% ethanol and 100% ethanol (79.78 and 55.15%, respectively).” 79.78% is not mentioned in the table 1, please recheck.

11.  Line 164-165, “It was also found that the DPPH radical scavenging activity of red radish extracts decreased with increasing water content in aqueous solvents.” By observing Table 1, the DPPH free radical scavenging activity of red radish treated with 70% methanol was 85.84%, but the data of red radish treated with 100% methanol was 76.86%. Didn't it increase with the increase of water content?

12.  Line 231, “250 mg/100 gm.” Please recheck the quantity unit.

13.   Line 250-253, “Obtained results revealed that in some cases, as the concentration of water in ethanol or methanol decreases, the total flavonoids content of the extract increases, as observed for 100% ethanol or methanol extract.” However, with the decrease of the concentration of water in ethanol or methanol, the total flavonoids content in the extracts of white radish and carrot decreases. Please rephrase this trend.

14.  For Antioxidant activity of polyphenols, you can cite these references ()

(a) Raghavan, S., & Kristinsson, H. G. (2008). Antioxidative efficacy of alkali-treated tilapia protein hydrolysates: a comparative study of five enzymes. Journal of Agriculture and Food Chemistry, 56(4), 1434-1441.

(b) Gao, J., Mao, Y. Z., Xiang, C. Y., Cao, M. N., Ren, G. R., Xie, H. J., Wang, K. W., Ma, X. J., Wu, D. (2021) Preparation of β-lactoglobulin/gum arabic complex nanoparticles for encapsulation and controlled release of EGCG in simulated gastrointestinal digestion model. Food Chemistry. 2021, 354, 129516.

(c) Rosa, A., Atzeri, A., Deiana, M., Melis, M.P., Incani, A., Minassi, A., et al. (2014). Prenylation preserves antioxidant properties and effect on cell viability of the natural dietary phenol curcumin. Food Research International, 57, 225–233.

15.  Line 276-277, “Similar observations were also clearly observed in figure 1 d, which also showed that 70% ethanol is the most valid and optimum extract for the aerial part of the carrot plant.” According to the expression at line 268, carrots are shown in Fig 1 e. Please rewrite.

Author Response

Responses to the reviewer’s comments

Reviewer 3

Dear Sir:

Thank you very much for your evaluation and valuable comments on our manuscript (agriculture-1883605). In response to your kind and helpful suggestions, we have revised our manuscript and the changes we made are highlighted in red in the manuscript text. Your comments, the revisions made, and our responses to your remarks are listed below:

Reviewer 3

Hassan et al. reported the effects of extraction solvents on total phenolic content, total flavonoid content, and antioxidant activity in the aerial part of root vegetables. This work is interesting and it would be better after the following issues are solved.

  1. What is the innovation of this article?

Response:  These non-edible parts considered as one of the wasted food. Estimation the antioxidant capacity of wasted part is required, so as to explore the potentials as alternative sources of antioxidants with added value for cosmetic, pharmaceutical, or food industrial applications

https://doi.org/10.1016/B978-0-323-85253-1.00020-4

https://doi.org/10.1016/B978-0-12-813768-0.00002-5

  1. When the dried aerial parts of the root vegetables are treated with ethanol and methanol, whether the solvent removal step is included, and whether it is safe to add organic solvents?

Response: yes, after extraction the solvents were removed and the extract was dried.  

  1. Line 80-85, as for the DPPH scavenging assay, the number of mL of DPPH solution added to the samples and the concentration of DPPH solution are not mentioned. It is suggested to supplement and improve DPPH experimental method.

Response:  Done

  1. As for the DPPH scavenging assay, what is the purpose of adding a 50 mm Tris-HCl buffer (pH 7.4)?

Response:  The Tris-HCl (pH 7.4) used in this standard method as catalyst.

  1. Line 90-92, “Briefly, 2.5 mL of phosphate buffer (0.2 M, pH 6.6) and 2.5 mL of 1% potassium ferricyanide were added to different concentrations of extracts (5, 10, 20, 30, and 40 μg/mL)”. How many mL of extract samples with different concentrations? Please add in the text.

Response:  corrected

  1. Line 101, “a phosphate buffer (pH 7.4)”. What is the phosphate buffer concentration?

Response:  added

  1. Line 120-121, “1 N sodium hydroxide (2 mL) was added to the mixture”. What is 1 N?

Response: concentration of the NaOH  

  1. Line 141-143, “with a descending manner of activity as follows: 70% ethanol > 100% methanol > 70% methanol > 100% ethanol > water.” In terms of significance, 100% methanol and 70% methanol, 100% ethanol and water were not significant.

Response:  yes this what exactly described in table 1.

  1. Line 145-146, “The 70% methanolic extract had a significantly (p < 0.05) higher DPPH radical scavenging activity (85.84%), followed by the 70% ethanolic extract (81.43%), 100% ethanolic extract (54.41%), and the aqueous extract (40.46%).” By observing Table 1, except that the aqueous extract (40.46%) is the data of white radish, others are the data of red radish. Please recheck the accuracy of the data.

Response:  Thank you for this observation. corrected

  1. Line 151-153, “When 70% methanol was used, carrot extract exhibited significantly higher scavenging activity (82.53%, P < 0.05) which was followed by 70% ethanol and 100% ethanol (79.78 and 55.15%, respectively).” 79.78% is not mentioned in the table 1, please recheck.

Response:  corrected

  1. Line 164-165, “It was also found that the DPPH radical scavenging activity of red radish extracts decreased with increasing water content in aqueous solvents.” By observing Table 1, the DPPH free radical scavenging activity of red radish treated with 70% methanol was 85.84%, but the data of red radish treated with 100% methanol was 76.86%. Didn't it increase with the increase of water content?

Response:  if you check in table 1 it was clearly written, for both ethanol and methanol, the absolute solvent (100%) has the lower value of DPPH.

  1. Line 231, “250 mg/100 gm.” Please recheck the quantity unit.

Response:  corrected

  1. Line 250-253, “Obtained results revealed that in some cases, as the concentration of water in ethanol or methanol decreases, the total flavonoids content of the extract increases, as observed for 100% ethanol or methanol extract.” However, with the decrease of the concentration of water in ethanol or methanol, the total flavonoids content in the extracts of white radish and carrot decreases. Please rephrase this trend.

Response:  corrected.

  1. For Antioxidant activity of polyphenols, you can cite these references ()

(a) Raghavan, S., & Kristinsson, H. G. (2008). Antioxidative efficacy of alkali-treated tilapia protein hydrolysates: a comparative study of five enzymes. Journal of Agriculture and Food Chemistry, 56(4), 1434-1441.

(b) Gao, J., Mao, Y. Z., Xiang, C. Y., Cao, M. N., Ren, G. R., Xie, H. J., Wang, K. W., Ma, X. J., Wu, D. (2021) Preparation of β-lactoglobulin/gum arabic complex nanoparticles for encapsulation and controlled release of EGCG in simulated gastrointestinal digestion model. Food Chemistry. 2021, 354, 129516.

(c) Rosa, A., Atzeri, A., Deiana, M., Melis, M.P., Incani, A., Minassi, A., et al. (2014). Prenylation preserves antioxidant properties and effect on cell viability of the natural dietary phenol curcumin. Food Research International, 57, 225–233.

Response:  Thank you for suggesting these interested references, however, I found that all these references are not fit with the aim of this manuscript.

  1. Line 276-277, “Similar observations were also clearly observed in figure 1 d, which also showed that 70% ethanol is the most valid and optimum extract for the aerial part of the carrot plant.” According to the expression at line 268, carrots are shown in Fig 1 e. Please rewrite.

Response:  corrected.

Thank you very much

Sincerely Yours

Dr. Amro B. Hassan

The Corresponding author

Round 2

Reviewer 2 Report

Kindly find below some few suggestions:

line 22: ...'optimum solvent for extraction of ...' instead of 'optimum extract for extraction'

line 70: antioxidant (small case)

line 71: indicate period samples were obtained

line 90: Which instrument/specification was used to determine absorbance?

line 99? report as- 1038 x g for 10 min (indicate temperature and centrifuge specification)

line 148: 'p' should be in small case

line 157: check repeated statement

(80.49%)100% ethanolic extract (54.41%), and lastly 100 ethanolic extract (54.41%).

line 174 to 175: statement made is not accurate;

methanol and ethanol with 30% water had higher antioxidant activity than methanol and ethanol (100%) with no water, So DPPH activity increased with increasing water content of organic solvents in the case of red raddish

Author Response

Responses to the reviewer’s comments

Dear Sir

Thank you very much for your evaluation and valuable comments on our manuscript (agriculture-1883605). In response to your kind and helpful suggestions, we have revised our manuscript and the changes we made are highlighted in red in the manuscript text. Your comments, the revisions made, and our responses to your remarks are listed below:

line 22: ...'optimum solvent for extraction of ...' instead of 'optimum extract for extraction'

Response: Done

line 70: antioxidant (small case)

Response: Done

line 71: indicate period samples were obtained

 Response: Done

line 90: Which instrument/specification was used to determine absorbance?

Response: Done

line 99? report as- 1038 x g for 10 min (indicate temperature and centrifuge specification)

Response: Done

line 148: 'p' should be in small case

Response: Done

line 157: check repeated statement

(80.49%)100% ethanolic extract (54.41%), and lastly 100 ethanolic extract (54.41%).

Response: Done

line 174 to 175: statement made is not accurate;

methanol and ethanol with 30% water had higher antioxidant activity than methanol and ethanol (100%) with no water, So DPPH activity increased with increasing water content of organic solvents in the case of red raddish

Response: Done

Sincerely Yours

Dr. Amro B. Hassan

The Corresponding author
